# Tunnel Surface Settlement Forecasting with Ensemble Learning

**Ke Yan [1,\*], Yuting Dai [2], Meiling Xu [3] and Yuchang Mo [4]**

[1]  Department of Building, School of Design and Environment, National University of Singapore, Architecture Drive, Singapore 117566, Singapore

[2]  Zhejiang Geely Holding Group Co., LTD. 1760, Jiangling Road, Binjiang District, Hangzhou 310051, China; daiyuting31@gmail.com

[3]  Nanhu College, Jiaxing University, Jiaxing 314001, China; meilingxunh@gmail.com

[4]  Fujian Province University Key Laboratory of Computational Science, School of Mathematical Sciences, Huaqiao University, Quanzhou 362021, China; myc@hqu.edu.cn

\*  Correspondence: yanke@nus.edu.sg; Tel.: +65-9875-5205

**Abstract:** Ground surface settlement forecasting in the process of tunnel construction is one of the most important techniques towards sustainable city development and preventing serious damages, such as landscape collapse. It is evident that modern artificial intelligence (AI) models, such as artificial neural network, extreme learning machine, and support vector regression, are capable of providing reliable forecasting results for tunnel surface settlement. However, two limitations exist for the current forecasting techniques. First, the data provided by the construction company are usually univariate (i.e., containing only the settlement data). Second, the demand of tunnel surface settlement is immediate after the construction process begins. The number of training data samples is limited. Targeting at the above two limitations, in this study, a novel ensemble machine learning model is proposed to forecast tunnel surface settlement using univariate short period of real-world tunnel settlement data. The proposed Adaboost.RT framework fully utilizes existing data points with three base machine learning models and iteratively updates hyperparameters using current surface point locations. Experimental results show that compared with existing machine learning techniques and algorithms, the proposed ensemble learning method provides a higher prediction accuracy with acceptable computational efficiency.

**Keywords:** tunnel settlement; time series analysis; ensemble learning; Adaboost.RT algorithm

## 1. Introduction

Modern sustainable city development involves various tunnel constructions, such as subway rail train tunnels, city underpass tunnels, and highway tunnels. Ground settlement in the process of tunnel construction is inevitable. Tunnel settlement not only affects the development of urban rail transit, it is also a great threat to the safety of the lives and property of urban area residents [1]. Data-driven forecasting method of the tunnel ground surface settlement is helpful to prevent serious damages and also useful for sustainable usage of the constructed tunnels [2–4].

Machine learning (ML), representing one of the hottest topics in data-driven methods, has achieved great successes in recent research studies. Compared to model-based methods, which require expert knowledge and experience to build physical or mathematical models, ML methods predict the future tunnel surface settlement purely based on historical data [5]. The established ML models are usually much more complex than the conventional mathematical model and can hardly be interpreted using several mathematical equations. The advantages of ML methods are easy for implementation and direct

for real-world application usage. Moreover, since the models are more complex, the forecasting accuracy usually can achieve a relatively high level and outperform most of the existing model-based methods.

In 2019, Hu et al. [5] compared most existing ML techniques for ground surface settlement prediction in real-world tunnel construction processes and pointed out that deep learning techniques, such as convolutional neural networks (CNNs) and long short-term memory (LSTM) neural networks, are not suitable for analyzing short sequence time series data, such as tunnel settlement data [6]. This study is a follow-up work of [5]. There are two main difficulties for tunnel settlement forecasting. First, the lengths of the historical data of tunnel constructions are limited. Second, the data collected are usually univariate. Deep learning methods usually are not suitable under the above-mentioned situations [7]. In contrast, traditional ML techniques are more reliable prediction methods for tunnel settlement in the short-term. In a study by Hu et al. [5], through a series of comparative study, three machine learning techniques, namely, back-propagation neural network (BPNN) [8], extreme learning machine (ELM) [9], and support vector regression (SVR) [10], were selected as best predicting the tunnel settlement in the short-term for various cases. Following this study [5] and using the same dataset, in this study, we show that there exists a complex nonlinear relationship between tunnel settlement and many random uncertain factors; it is difficult to predict the tunnel settlements using one single machine learning technique.

In this research, an ensemble learning approach that integrates multiple machine learning models is proposed. To verify the performance of the method proposed, a real-world tunnel surface settlement from a tunnel construction corporation is selected. The motivation of this study is searching for the most suitable data-driven forecasting technique for short-term tunnel surface settlement. The study makes the following contributions to the literature.

(1) Utilizing multiple machine learning models for improving the tunnel settlement prediction accuracy: Based on the literature review conducted in Section 2, using one single model to predict the tunnel settlement accurately is indeed challenging. In this study, multiple different types of models, such as SVR, BPNN, and ELM, were integrated to construct a powerful ensemble learning framework for tunnel settlement forecasting.

(2) A customized ensemble learning algorithm based on traditional Adaboost.RT algorithm: The traditional Adaboost.RT algorithm was altered to best suit the short-term forecasting with a limited number of training data samples from the ground sensors during the tunnel construction period. The traditional Adaboost.RT algorithm usually integrates one type of prediction model for ensemble learning. In this study, we reimplement the Adaboost.RT algorithm integrating three types of different prediction models.

(3) Illustrating the prediction performance improvement through a comprehensive comparative study: In the experimental section, results are illustrated to compare the proposed method with existing works. First, we show that the ensemble learning framework outperforms individual base classifiers. Next, more experiments were carried out to compare the proposed framework with traditional ensemble learning approaches (i.e., the Adaboost.RT framework integrating a single type of classifiers). The performance of the proposed ensemble learning framework is proven through a series of comparative studies.

## 2. Related Works

Short-term time series data forecasting is a popular topic in the field of regression analysis, machine learning, and intelligent computing. In general, those research methods can be divided into two categories: model-based methods and data-driven methods [11,12]. The model-based methods include numerical analysis method, numerical simulation method, semi-theoretical analytical method, and stochastic theoretical model. For example, an empirical method based on the normal distribution function was proposed by Fang et al. to estimate the magnitude of tunnel surface settlement [13]. An elastic-visco-plastic (EVP) constitutive model in triaxial space and general stress space, used mainly for the study of isotropic clays, was proposed by Islam and Gnanendran [14]. They carried out many

experimental studies on Kaolin clay, Hong Kong marine deposit clay, and Fukakusa clay, and all experiments achieved good prediction results. Mei et al. [15] started with the relationship between time and settlement, and mathematically proved that the settlement curve under the linear load applied to construction and embankment engineering is an "S" shape. On this basis, a new settlement prediction model called 'Poison model', provides an effective predictive study of building and surface settlement.

The data-driven methods mainly refer to ML methods. Recent study shows that ML-based forecasting strategies usually provide extremely high prediction results for time series data forecasting problems [3,16]. Ocak and Seker [17] compared artificial neural network (ANN), support vector machine (SVM), and Gaussian processes (GPs) on surface settlement forecasting calculation caused by earth pressure balance machines (EPBMs). The experiment results show that SVM has the best performance. Azadi and Pourakbar [18] used the finite element method to study the settlement of buildings around the tunnel, and then used neural network to analyze various settlements, and finally obtained the prediction conclusion of tunnel settlement. Moghaddasi et al. [19] utilized an imperialist competitive algorithm (ICA)-enhanced ANN algorithm to predict the tunnel settlement and achieved outstanding prediction results.

Ensemble or hybrid machine learning techniques are important solutions to raise the prediction accuracy of traditional machine learning techniques [20–24]. Tang et al. [25] proposed a hybrid ensemble learning framework to forecast nuclear energy consumption patterns. The experimental results show that the hybrid ensemble learning framework outperforms the single LSSVR learning method. Li et al. [26] introduced a wavelet-based ensemble learning framework for the short-term load forecasting problem. Wavelet transform was combined with multiple ELMs to boost the forecasting performance. Wang et al. [27] combined wavelet transform with convolutional neural network to forecast wind power data as time series data. The results showed that the volatility and irregularity of the wind power can be adaptively learned by the proposed hybrid learning method. In the current context of machine learning technology, the most popular ensemble learning algorithm is Adaboost and its extensions, including Adaboost.M1, Adaboost.M2 [28], Adaboost.R, Adaboost.R2 [29], and Adaboost.RT [30] algorithms.

## 3. Materials and Methods

Different from traditional forecasting problems, tunnel surface settlement forecasting has the challenges/properties of a short-period of time data available, univariate training data, and various hidden factors that are missing in the dataset. Existing works, such as [5], have shown the instability of prediction models while only single type models are used. To improve the robustness of the final prediction model, an integration of multiple models using ensemble learning algorithms is desired.

### 3.1. Selection of Base Prediction Models

In order to get an ensemble learner with better generalization performance and high prediction accuracy, the individual base prediction models have to be effective themselves and independent from each other. Nevertheless, completely independent models are hard to fit in actual tasks. For traditional Adaboost algorithms the same type of base classifiers is utilized. The base models differ from each other following different sample distribution in the training process. In this study, the original Adaboost.RT algorithm was altered by selecting different types of base learners to obtain the final ensemble generalized learning framework. According to our previous study on this topic [5], three base classifiers, namely, BPNN, ELM, and SVR were selected to build the ensemble learner.

Among the three base classifiers, SVR has strong generalization ability, which can easily solve small sample, over-learning, high-dimensional, and local minimum problems. BPNN has strong adaptive abilities, self-organizing, self-learning, and non-linear mapping characteristics. It overcomes the shortcomings of traditional feedback methods and has been used more widely in recent years. The ELM is a single-layer neural network, which has a fast convergence rate, high prediction accuracy, and strong generalization properties.

*3.2. The Proposed Method*

The proposed Adaboost.RT algorithm is an improved algorithm based on Adaboost.R2 by Solomatine and Shrestha [18]. They introduced a threshold in the Adaboost algorithm and compared the training error of every sample with the threshold, after that the training set was divided into two categories, and the regression problem was translated into a simple two-class problem, which can be dealt with by the traditional AdaBoost algorithm. In the process of converting a regression problem into a binary classification problem, AdaBoost.RT employs the absolute relative error (ARE) as an error metric to differentiate samples into true/false predictions; while the ARE value of one testing sample is higher than the threshold $\phi$, the last predictor is recognized as the unsuitable prediction model and the tested sample will be tested again using other prediction models. The traditional Adaboost.RT algorithm integrates multiple base predictors of the same type and the threshold value $\phi$ is pre-set and manually adjusted.

In this study, we altered the traditional Adaboost.RT algorithm using three different types of base prediction models, namely, SVR, BPNN, and ELM. The initial weight or distribution of each data sample was equivalent. If there are $m$ training data samples, the initial weight of each sample is $1/m$. The distribution of each data sample was updated after evaluating the prediction result of SVM using the threshold value $\phi$. The distributions were updated two more times for BPNN and ELM. The threshold value $\phi$ of Adaboost.RT was adaptively calculated instead of pre-setting using Zhang and Yang's method [31].

The customized Adaboost.RT algorithm is elaborated in Algorithm 1. The overall flowchart is depicted in Figure 1.

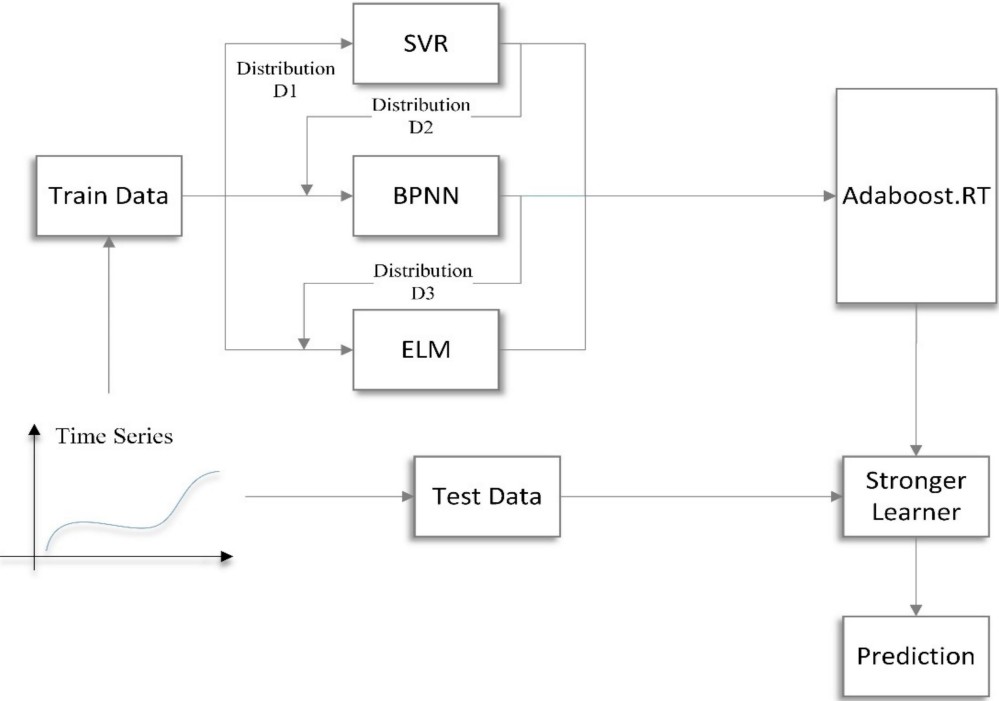

**Figure 1.** Overall flowchart of the customized Adaboost.RT algorithm integrating multiple models. Abbreviations: SVR, support vector regression; BPNN, back-propagation neural network; ELM, extreme learning machine.

---

**Algorithm 1.** A customized Adaboost.RT algorithm for tunnel settlement forecasting

---

**Input**: Training dataset $M$, weak learning algorithm (base learner) $l$, integer $T$ specifying number of iterations (machines), threshold $\phi$ for differentiating correct, and incorrect predictions.

**Initialization:** Error rate $\varepsilon_t$, sample distribution $D_t(i) = 1/m$, machine number or iteration $t = 1$.

**Iteration**: While $t < T$:

**Step 1:** Calling base learner, providing it with distribution $D_t(i) = 1/m$.

**Step 2:** Building a regression model:

$$f_t(x) \rightarrow y$$

**Step 3:** Calculating absolute relative error for each training example as

$$ARE_t(i) = \left| \frac{f_t(x_i) - y_i}{y_i} \right|$$

**Step 4:** Calculating the error rate:

$$f_t(x) : \varepsilon_t = \sum_{i:ARE_t(i)>\phi} D_t(i)$$

**Step 5:** Setting $\beta_t = (\varepsilon_t)^h$, where $h = 1, 2$ or $3$ (linear, square or cubic).

**Step 6:** Updating distribution $D_t(i)$

$$D_{t+1}(i) = \frac{D_t(i)}{Z_t} \times \begin{cases} \beta_t, if\ ARE_t(i) \le \phi \\ 1, otherwise \end{cases}$$

where $Z_t$ is a normalization factor chosen such that $D_{t+1}$ will be a distribution.

**Step 7:** Adjusting $\phi$ according to the algorithm proposed in [31].

**Step 8:** Setting $t = t + 1$

**Output:** Outputting the ensemble learner:

$$f_{ensemble}(x) = \frac{\sum\limits_{t=1}^{T} \log(\frac{1}{\beta_t}) f_t(x)}{\sum\limits_{t=1}^{T} \log(\frac{1}{\beta_t})}$$

---

## 4. Results

A real-world dataset that was collected by a tunnel construction company located in Zhuhai City, China in 2015 was used. The data was recorded in chronological order for each collection point of the subway tunnel construction (attached as Supplementary File: Table S1). It is a single-dimension time series dataset. The tunnel ground surface points are indexed from 180 to 250 (see Supplementary Table S1). Among the 70 ground surface points, nine representative points are selected along the tunnel medial axis. The selected points are indexed: 181, 182, 184, 188, 189, 190, 210, 225, and 235.

Since the sample data is a one-dimension time series, one problem is that the sample dimension is too low. A suitable rolling window size was selected to expand the original single-dimensional data into multi-dimension data [5]. According to the dataset scales and experience, the best rolling window size is between 1 and 20. In the actual experiments, the rolling window size finally lied at 3. Then the original single-dimension sample data were expanded into three-dimension sample data, thereby solving the problem of too low dimension. In addition, for each tunnel ground surface point, 5/6 of the records was utilized as training dataset. The remaining 1/6 of the records was treated as testing data for verification purposes.

The data of subway tunnel surface settlement records from Zhuhai rail transit were selected for simulation, and the threshold value was adjusted by bisectional method using multiple training processes to obtain the final results.

To show the superiority of the performance of the proposed Adaboost.RT framework integrating multiple models, SVR, BPNN, and ELM were also developed for predicting the tunnel surface settlement. Figures 2–7 show the forecasting results for surface point numbers 181, 184, 188, 189, 201, and 220. The testing results are enlarged in subfigures.

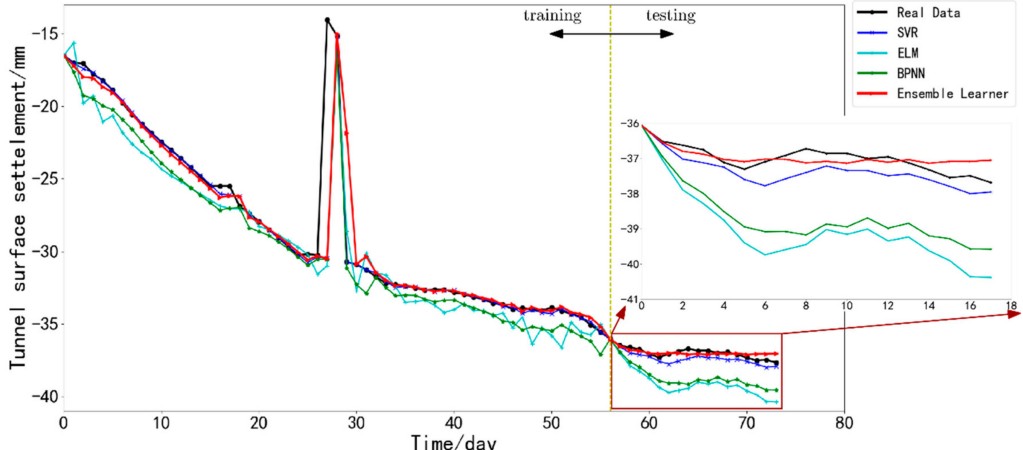

**Figure 2.** Prediction results of tunnel surface point number 181. The testing results are enlarged in the subfigure.

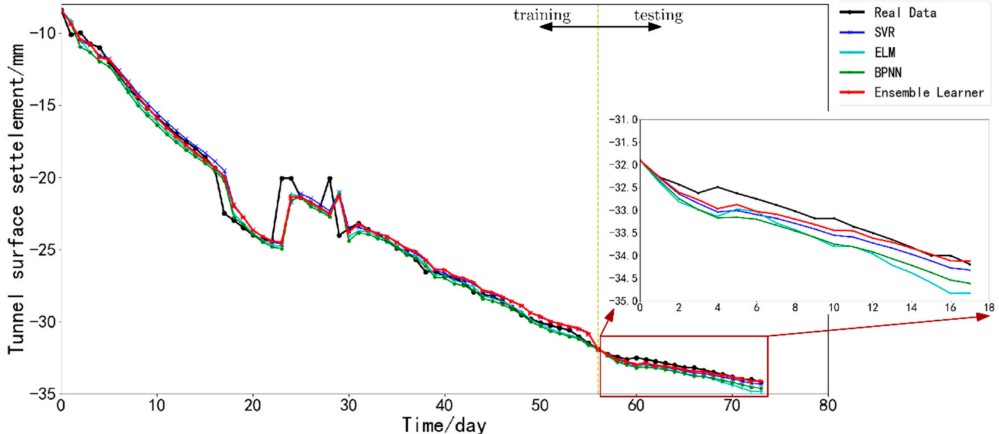

**Figure 3.** Prediction results of tunnel surface point number 184.

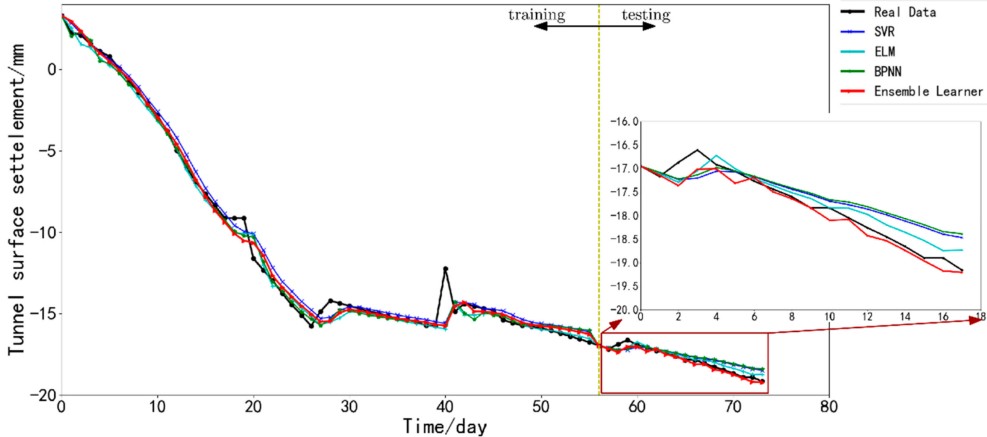

**Figure 4.** Prediction results of tunnel surface point number 188.

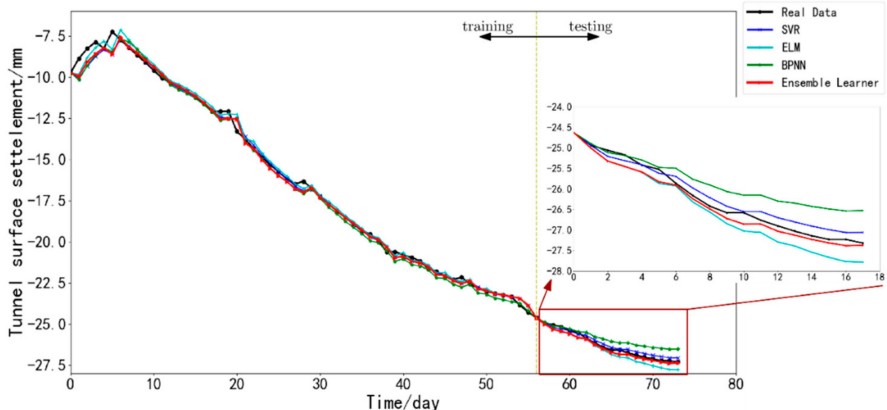

**Figure 5.** Prediction results of tunnel surface point number 189.

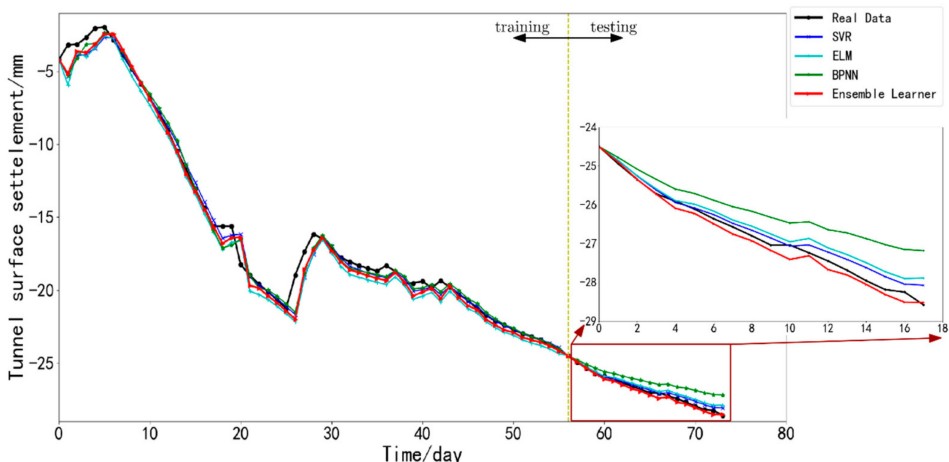

**Figure 6.** Prediction results of tunnel surface point number 201.

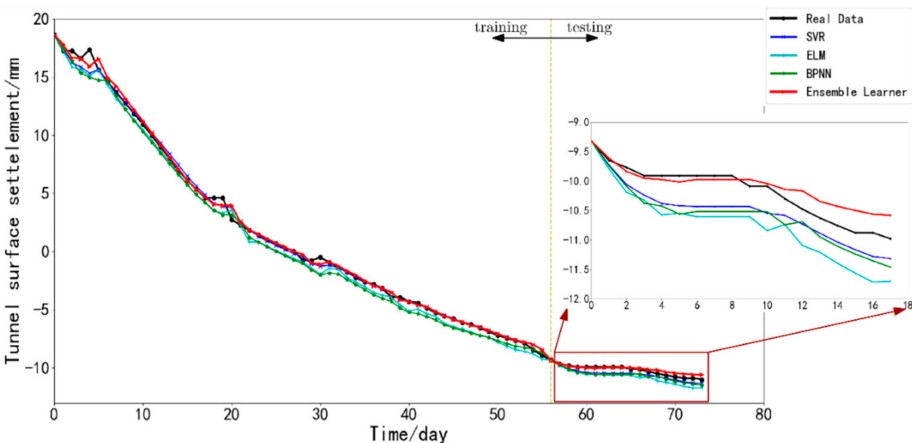

**Figure 7.** Prediction results of tunnel surface point number 220.

The testing computer's configuration consists of an Intel Core (TM) i7-8700K CPU @ 3.70 GHz, NVIDIA GeForce GTX1080 graphics card, 16 GB RAM, and 8 GB graphical memory with Python version 3.7 (64-bit) and Keras version 2.0.3. Since all tested methods are machine learning models each test was finished within one minute.

All results are shown in different colors. The results of the proposed ensemble learning algorithm are shown in red and the actual data points are shown in black. The prediction results of the rest of the compared methods are shown in other colors. It can be clearly seen from the figures that the method of integrating multiple models with Adaboost.RT has an obviously high prediction accuracy.

The performance of each method was evaluated by three error measurement metrics, namely, root mean squared error (RMSE), mean absolute error (MAE), and mean absolute percentage error (MAPE). The formulas of the three error metrics are listed in Equations (1)–(3):

$$RMSE = \sqrt{\frac{1}{n}\sum_{i=1}^{n}(y_i - y_i')^2} \tag{1}$$

$$MAE = \frac{1}{n}\sum_{i=1}^{n}\left|y_i' - y_i\right| \tag{2}$$

$$MAPE = \frac{1}{n}\sum_{i=1}^{n}\left|\frac{y_i' - y_i}{y_i}\right| \times 100\% \tag{3}$$

where $y'$ is the predicted value and $y$ is the actual value. The results of the three errors metrics (RMSE, MAE, and MAPE) of different methods for tunnel surface settlement are listed in Table 1. For all cases, the Adaboost.RT integrating multiple models had the best RMSE, MAE, and MAPE values according to Table 1.

In this research, to further validate the performance and robustness of the proposed method, a comparative experiment with traditional Adaboost.RT models integrating one type (SVR, BP or ELM) of classifier was conducted (Table 2). For example, the Adaboost.RT + SVR classifier is built by integrating three SVRs in the Adaboost.RT framework. In Table 2, the compared methods were also trained using the same sets of data to get the final ensemble learner. The results show that the proposed extended Adaboost.RT algorithm outperforms the traditional Adaboost.RT models integrating one type of classifier.

From Tables 1 and 2, it is evident that the Adaboost.RT integrating multiple models method has better performance compared with the rest of the compared methods. The reason for this is because there exists a nonlinear relationship between time and tunnel settlement data, which can hardly be captured by one single machine learning technique. In summary, the proposed Adaboost.RT algorithm is a more appropriate method for tunnel surface settlement forecasting in the short-term compared to traditional predictive models.

**Table 1.** Prediction results of all tunnel surface point settlements by different methods.

| Point | Proposed | | | SVR | | | BPNN | | | ELM | | |
|---|---|---|---|---|---|---|---|---|---|---|---|---|
| | RMSE | MAE | MAPE (%) | RMSE | MAE | MAPE (%) | RMSE | MAE | MAPE (%) | RMSE | MAE | MAPE (%) |
| 181 | 0.2641 | 0.2041 | 0.5488 | 0.4158 | 0.3684 | 0.9950 | 1.7444 | 1.6349 | 4.4094 | 2.1531 | 2.0166 | 5.4360 |
| 182 | 0.2945 | 0.2470 | 1.2853 | 0.3675 | 0.3113 | 1.6157 | 0.3807 | 0.3168 | 1.6391 | 0.3067 | 0.2548 | 1.3304 |
| 184 | 0.2001 | 0.1587 | 0.4796 | 0.3960 | 0.3653 | 1.1017 | 0.4342 | 0.4027 | 1.2115 | 0.5547 | 0.5169 | 1.5559 |
| 188 | 0.1953 | 0.1370 | 0.7826 | 0.3773 | 0.3071 | 1.6968 | 0.4038 | 0.3261 | 1.7950 | 0.2463 | 0.1985 | 1.1090 |
| 189 | 0.1597 | 0.1325 | 0.5083 | 0.1702 | 0.1484 | 0.5587 | 0.4949 | 0.4084 | 1.5241 | 0.3200 | 0.2825 | 1.0639 |
| 190 | 0.1375 | 0.0867 | 0.3058 | 0.1610 | 0.1075 | 0.3793 | 0.1455 | 0.0973 | 0.3448 | 0.1650 | 0.1166 | 0.4132 |
| 210 | 0.1530 | 0.1239 | 0.4873 | 0.3358 | 0.2157 | 0.8560 | 0.2274 | 0.1741 | 0.6898 | 0.2254 | 0.1786 | 0.7066 |
| 225 | 0.1821 | 0.1473 | 0.3585 | 0.1986 | 0.1587 | 0.3862 | 0.2822 | 0.2132 | 0.5193 | 0.2624 | 0.1974 | 0.4808 |
| 235 | 0.2619 | 0.1979 | 0.9103 | 0.3779 | 0.3272 | 1.4988 | 0.5462 | 0.4614 | 2.1122 | 0.2867 | 0.2498 | 1.1457 |

Abbreviations: RMSE, root mean squared error; MAE, mean absolute error; MAPE, mean absolute percentage error.

**Table 2.** Prediction results of all tunnel surface point settlements by different Adaboost.RT algorithms.

| Point Number | Proposed | | | Adaboost.RT + SVR | | | Adaboost.RT + BPNN | | | Adaboost.RT + ELM | | |
|---|---|---|---|---|---|---|---|---|---|---|---|---|
| | RMSE | MAE | MAPE (%) | RMSE | MAE | MAPE (%) | RMSE | MAE | MAPE (%) | RMSE | MAE | MAPE (%) |
| 181 | 0.2641 | 0.2041 | 0.5488 | 0.3901 | 0.3395 | 0.9179 | 0.4201 | 0.3616 | 0.9723 | 1.2479 | 1.1167 | 3.0138 |
| 182 | 0.2945 | 0.2470 | 1.2853 | 0.3082 | 0.2645 | 1.3765 | 0.3019 | 0.2629 | 1.3661 | 0.3305 | 0.2862 | 1.4857 |
| 184 | 0.2001 | 0.1587 | 0.4796 | 0.1943 | 0.1618 | 0.4901 | 0.2392 | 0.1847 | 0.5615 | 0.4215 | 0.3913 | 1.1773 |
| 188 | 0.1953 | 0.1370 | 0.7826 | 0.2193 | 0.1569 | 0.8887 | 0.2981 | 0.2406 | 1.3338 | 0.1976 | 0.1475 | 0.8324 |
| 189 | 0.1597 | 0.1325 | 0.5083 | 0.2609 | 0.2131 | 0.8072 | 0.3616 | 0.2963 | 1.1064 | 0.3674 | 0.3031 | 1.1320 |
| 190 | 0.1375 | 0.0867 | 0.3058 | 0.1700 | 0.1168 | 0.4126 | 0.1569 | 0.1190 | 0.4271 | 0.1424 | 0.0933 | 0.3305 |
| 210 | 0.1530 | 0.1239 | 0.4873 | 0.2158 | 0.1829 | 0.7137 | 0.1884 | 0.1538 | 0.6063 | 0.1665 | 0.1410 | 0.5518 |
| 225 | 0.1821 | 0.1473 | 0.3585 | 0.1969 | 0.1565 | 0.3811 | 0.2090 | 0.1620 | 0.3945 | 0.2142 | 0.1649 | 0.4011 |
| 235 | 0.2619 | 0.1979 | 0.9103 | 0.3590 | 0.3061 | 1.3999 | 0.3327 | 0.2865 | 1.3114 | 0.2701 | 0.2362 | 1.0834 |

Abbreviations: RMSE, root mean squared error; MAE, mean absolute error; MAPE, mean absolute percentage error.

## 5. Conclusions, Limitations, and Future Works

This study proposed a novel extended Adaboost.RT algorithm to forecast ground surface settlement in tunnel construction processes. The proposed extended Adaboost.RT algorithm integrates different types of base classifiers instead of a single type of classifiers (as in the traditional way). The base classifiers were selected according our previous study on the same dataset. In the training phase, the Adaboost.RT algorithm was used to improve different type models by distribution that were constantly being updated.

Comprehensive experiment results were shown in Section 4 to demonstrate the effectiveness of the proposed ensemble learning framework. First, we compared the proposed framework with individual machine learning classifiers, including SVR, BPNN, and ELM. Next, more experiments were carried out to compare the proposed framework with traditional ensemble learning approaches, (i.e., the Adaboost.RT framework integrating a single type of classifiers). The performance of the proposed ensemble learning framework is the best among all compared methods.

The main limitation of this study is that only one tunnel settlement dataset is tested. The dataset, as we attached to this manuscript as a Supplementary File (Table S1), contains only 70 surface points data. Out of the 70, we selected nine representative points along the tunnel medial axis to show the prediction results of the proposed Adaboost.RT forecasting framework. As one of the future works, the proposed method will be applied to more real-world tunnel settlement dataset to verify the robustness. In addition, more extensions of the Adaboost algorithms will be implemented and evaluated.

**Supplementary Materials:** The following are available online at http://www.mdpi.com/2071-1050/12/1/232/s1. Table S1: Tunnel settlement data collected from Zhuhai subway construction project in China.

**Author Contributions:** Conceptualization, K.Y. and Y.D.; methodology, K.Y.; software, Y.D.; validation, K.Y., M.X. and Y.M.; formal analysis, Y.M.; investigation, K.Y.; resources, K.Y.; data curation, Y.M.; writing—original draft preparation, K.Y.; writing—review and editing, K.Y.; visualization, M.X.; supervision, K.Y.; project administration, K.Y.; funding acquisition, K.Y. All authors have read and agreed to the published version of the manuscript.

**Funding:** This work was supported the faculty start-up research grant of National University of Singapore under grant number R-296-000-208-133 (K.Y.), research project on the "13th Five-Year Plan" of higher education reform in Zhejiang Province, under grant number JG20180526 (M.X.), National Natural Science Foundation of China under trant 61972156, and Program for Innovative Research Team in Science and Technology in Fujian Province University (Y.M.).

**Acknowledgments:** The authors would like to thank Shanghai Tunnel Engineering Co., Ltd for providing the tunnel settlement data from Zhuhai subway construction project in China.

**Conflicts of Interest:** The authors declare that they have no conflict of interest.

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
