# Peer review of "Tunnel Surface Settlement Forecasting with Ensemble Learning"

_sustainability, doi:10.3390/su12010232_

Round 1
Reviewer 1 Report
In general, this manuscript is properly written, and the structure is clear. The applied data sets and the analysis results are convincing. However, I do have the following concerns.
The authors used the integration of SVM, BP and ELM as the ensemble learning framework. But I didn't see the reason why the combination of these three. What if a combination of SVM and BP or a combination of BP and ELM? It was written that this integration referring to Hu et al. 2019, but it might a bit far-fetching. Is it representative enough that the authors chose only 9 surface points? How to prove it?Author Response
In general, this manuscript is properly written, and the structure is clear. The applied data sets and the analysis results are convincing. However, I do have the following concerns.
The authors used the integration of SVM, BP and ELM as the ensemble learning framework. But I didn't see the reason why the combination of these three.
Reply: Thanks for the reviewer’s comment. This work follows Hu et al.’s work published in 2019 using the same dataset. In Hu et al.’s work, they actually performed a comparative study and selected SVM, BP and ELM as the best machine learning techniques for the tunnel settlement dataset. We have revised the introduction part to make the selection clearer:
“In 2019, Hu et al. [5] compared most existing ML techniques for ground surface settlement prediction in real-world tunnel construction processes and pointed out that deep learning techniques, such as convolutional neural networks (CNNs) and long short term memory (LSTM) neural networks, are not suitable for analyzing short sequence time series data, such as the tunnel settlement data. In contrast, traditional ML techniques are more reliable prediction methods the tunnel settlement in short-term. In [5], through a series of comparative study, three machine learning techniques, namely, support vector regression (SVR), back-propagation neural network (BPNN) and extreme learning machine (ELM), are selected best predicting the tunnel settlement in short-term for various cases.”
Modifications made to the manuscript are marked in red color.
Reference:
[Hu et al.] Hu, M., Li, W., Yan, K., Ji, Z., & Hu, H. (2019). Modern machine learning techniques for univariate tunnel settlement forecasting: A comparative study. Mathematical Problems in Engineering, 2019.
What if a combination of SVM and BP or a combination of BP and ELM?
Reply: The two combinations mentioned above are subsets of the proposed Adaboost.RT algorithm. The influences of SVM, BP or ELM are controlled by the threshold ϕ, which is a parameter auto-adjusted to optimized the ensemble learner performance. The two combinations mentioned above are special cases when ELM or SVM totally have no influence to the data distribution, i.e., the input and output distributions are the same. Through testing on different thresholds ϕ, the cases above are included in the final results with automated parameter tuning.
It was written that this integration referring to Hu et al. 2019, but it might a bit far-fetching.
Reply: Thanks for the reviewer’s comment. We stated in the revised version (paragraph 2, marked in red) that this study is a follow-up work of Hu et al. 2019. We share the same dataset. In fact, this manuscript’s corresponding author (Dr Ke Yan) is also the corresponding author of Hu et al. 2019.
Is it representative enough that the authors chose only 9 surface points? How to prove it?
Reply: According to the dataset attached as the supplementary file to the manuscript, the entire dataset contains 70 ground surface points for the tunnel construction in Zhuhai City, China. We selected 9 points along the tunnel medial axis, which have more representative movements among all tested points. The number “9” is selected based on the dataset size and the article length (by displaying all prediction results). In overall, we believe that by showing the outperformance of the proposed method on all selected 9 points, and the results listed in Hu et al. 2019, it is evident that the proposed method is suitable for tunnel surface forecasting on the given dataset. We agree with the reviewer that the robustness is yet to be proved since we only use one dataset. In the future works part (Section 5) of the revised version, we mentioned that we will continue our work and test on more real-world datasets to verify the robustness of the proposed Adaboost.RT framework.
The first paragraph of Section 4 is revised and as well as the conclusion part in Section 5. All modifications made are marked in red.

Reviewer 2 Report
In the manuscript, the authors are trying to propose a new algorithm to address tunnel surface settlement forecasting issues. The manuscript is slightly dis-organized with too much unsorted background information while too short results presented. Thus, this manuscript should not be considered as a potential publication. Here provides some critical points:
In page 2, second paragraph from "In 2019,...." In this paragraph, the authors have introduced several ML techniques that ever be explored by others and claimed that they are not feasible to the issues in this manuscript. However, no detailed reasons provided. The first 2 sections plus 3.1, 3.2 are all illustrating previous works. Can these contents be combined into 1 or 2 sections as background information? It's recommended to include a table stating each ML techniques and their pros and cons, so that the motivation of this manuscript could also be added. Meanwhile, the correlation between previous works and this work should also be provided. For the last part of results, the authors just piled all figures and tables in the manuscript. It is required to reorganize all the plots. Since all the plots have same data format/range, could they be combined to have better comparisons? Although, the final error from current algorithm is lowest comparing with other techniques, the detailed reasons are not provided. In addition, the time complexity and memory usage of the algorithm have not been discussed.Author Response
In the manuscript, the authors are trying to propose a new algorithm to address tunnel surface settlement forecasting issues. The manuscript is slightly dis-organized with too much unsorted background information while too short results presented. Thus, this manuscript should not be considered as a potential publication. Here provides some critical points:
In page 2, second paragraph from "In 2019,...." In this paragraph, the authors have introduced several ML techniques that ever be explored by others and claimed that they are not feasible to the issues in this manuscript. However, no detailed reasons provided.
Reply: Thanks for the reviewer’s comment. This work is a follow-up work of [1]. Basically, the tunnel time series data is too short for deep learning. In fact, we have tried to use LSTM and its various extensions to forecast tunnel settlements. The performances of LSTM and its extension are not comparable to BPNN, SVR and ELM results, since the dataset that we use is really small (we have upload the dataset to the MDPI submission system). The reasoning was also stated clearly in [1] (which is why we did not state again in the first version of this paper). In the revised version, we have added some detailed reasoning (as explained above) to the Introduction section (paragraph 2, marked in red).
Reference:
[1] Hu, M., Li, W., Yan, K., Ji, Z., & Hu, H. (2019). Modern machine learning techniques for univariate tunnel settlement forecasting: A comparative study. Mathematical Problems in Engineering, 2019.
The first 2 sections plus 3.1, 3.2 are all illustrating previous works. Can these contents be combined into 1 or 2 sections as background information? It's recommended to include a table stating each ML techniques and their pros and cons, so that the motivation of this manuscript could also be added.
Reply: Thanks for the reviewer’s nice comment. In the revised version of the manuscript, we have moved part of the contents of Section 3, 3.1 and 3.2 to Section 2. In addition, some redundant and repeated contents have been removed. More literature reviews have been deleted to keep the background paragraphs short. The motivation of the manuscript has been emphasized in the last paragraph of the Introduction section (marked in red):
“The motivation of this study is searching for the most suitable data-driven forecasting technique for short-term tunnel surface settlement.”
Meanwhile, the correlation between previous works and this work should also be provided.
Reply: Thanks for the reviewer’s nice comment. In paragraph 2, we have added the statement saying that it is a follow-up work of [1] and we share the same dataset (marked in red).
For the last part of results, the authors just piled all figures and tables in the manuscript. It is required to reorganize all the plots. Since all the plots have same data format/range, could they be combined to have better comparisons?
Reply: We have re-draw Figures from 2 to 7 for better visualization effect on the manuscript. Since there are many lines and points to show in each figure, it is difficult to merge multiple figures into one single figure. In fact, another reviewer has already commented that it is hard to distinguish the lines. And we have to enlarge the details of the testing part and form a sub-figure for each of the Figures 2-7.
Although, the final error from current algorithm is lowest comparing with other techniques, the detailed reasons are not provided.
Reply: Thanks. In the revised version, we have added more elaboration on the results listed in Table 1 and 2, Figures 2-7. Generally speaking, we believe that there exists a complex nonlinear relationship between tunnel settlement and many random uncertain factors, it is difficult to predict the tunnel settlements using one single machine learning technique.
In addition, the time complexity and memory usage of the algorithm have not been discussed.
Reply: We have added the configuration of the tested machine into the article (lines 213-216 on page 6): “The testing computer’s configuration consists of an Intel Core (TM) i7-8700K CPU @ 3.70 GHz, NVIDIA GeForce GTX1080 graphics card, 16GB RAM and 8GB graphical memory with Python version 3.7 (64-bit) and keras version 2.0.3. Since all tested methods are machine learning models. Each test is finished within one minute. ”
Since all methods we tested are machine learning models (not involving any deep learning techniques), the test runs are all fast. Each test result can be obtained within one minute.
The modifications made are marked in red.

Reviewer 3 Report
Paper on Tunnel Surface Settlement Forecasting by authors is interesting, and nicely written in most part. Some of the observations:
abbreviations are a bit messy: ELM, SVN, BPNN, ML, and SVR are defined multiple times; RMSE, MAE and MAPE appear before the explanation. readability of the figures could be improved in printed form. Text on axes is too small and lines are hard to differentiate in black and white. Some spelling errors are present, such as "token" on line 220. Generally sections 4 and 5 could be improved. "Comprehensive experiment results were shown in Section 5" - line 282 - Where in section 5?Author Response
Paper on Tunnel Surface Settlement Forecasting by authors is interesting, and nicely written in most part. Some of the observations:
Abbreviations are a bit messy: ELM, SVN, BPNN, ML, and SVR are defined multiple times; RMSE, MAE and MAPE appear before the explanation.
Reply: Thanks for the reviewer’s comment. We have removed all multi-definitions of the abbreviations of ELM, SVN, BPNN, ML, and SVR. The introduction paragraph of RMSE, MAE and MAPE has been revised. We carefully marked all modifications made to the revised version in red color.
Readability of the figures could be improved in printed form. Text on axes is too small and lines are hard to differentiate in black and white.
Reply: We have re-draw Figures 2-7 to improve the readability of the Figures. In particular, we have enlarged the texts on axes and bolded the lines to better distinguish different methods. Although we agree with the reviewer that the forecasting results such as those lines listed in Figures 2-7 are hard to be distinguished in black and white, we would like to clarify that, by paying an article processing charge (AFC) with MDPI, all papers will be freely available online in a colored version.
Some spelling errors are present, such as "token" on line 220.
Reply: Thanks for the reviewer’s nice comment. Yes. the word “token” is a typo, which should be “taken”. We have revised the sentence in line 220 (marked in red). In addition, in this round of revision, we have carefully proof-read the article to minimize the possibility of other typos.
Generally sections 4 and 5 could be improved.
Reply: Thanks for the reviewer’s nice comment. We have added more elaborations of the experimental results, conclusions and limitations to Section 4 and 5. All modifications made are marked in red in the revised version.
"Comprehensive experiment results were shown in Section 5" - line 282 - Where in section 5?
Reply: Thanks for the reviewer’s nice comment. The phrase “Section 5” is a typo, which should be “Section 4”.

Round 2
Reviewer 2 Report
The authors have addressed all the technical concerns from the reviewer, thus I would suggest consideration of the manuscript as potential publication.